

# Explanation of observational data engenders a causal belief about smoking and cancer

Leslie Myint[1], Jeffrey T. Leek[2] and Leah R. Jager[2]

[1] Department of Mathematics, Statistics, and Computer Science, Macalester College, St. Paul, MN, United States of America
[2] Department of Biostatistics, Johns Hopkins Bloomberg School of Public Health, Baltimore, MD, United States of America

## ABSTRACT

Most researchers do not deliberately claim causal results in an observational study. But do we lead our readers to draw a causal conclusion unintentionally by explaining why significant correlations and relationships may exist? Here we perform a randomized controlled experiment in a massive open online course run in 2013 that teaches data analysis concepts to test the hypothesis that explaining an analysis will lead readers to interpret an inferential analysis as causal. We test this hypothesis with a single example of an observational study on the relationship between smoking and cancer. We show that adding an explanation to the description of an inferential analysis leads to a 15.2% increase in readers interpreting the analysis as causal (95% confidence interval for difference in two proportions: 12.8%–17.5%). We then replicate this finding in a second large scale massive open online course. Nearly every scientific study, regardless of the study design, includes an explanation for observed effects. Our results suggest that these explanations may be misleading to the audience of these data analyses and that qualification of explanations could be a useful avenue of exploration in future research to counteract the problem. Our results invite many opportunities for further research to broaden the scope of these findings beyond the single smoking-cancer example examined here.

# ARTICLE

Drinking too much tea causes prostate cancer (*PTI, 2016*), eating chocolate helps people stay thin (*Jaslow, 2012*), junk food during pregnancy leaves children fat for life (*Mirror.co.uk, 2008*). We all know that correlation does not imply causation, but we have also all seen exaggerated headlines in the media that fall short in capturing the true results of a scientific study. A recent report in the British Medical Journal found the fault may not lie entirely with the media (*Sumner et al., 2014*), but may be aided by exaggerated press releases from universities themselves. In fact, in their study of 462 press releases, the study authors found that 33% (95% confidence interval: 26% to 40%) contained exaggerated causal claims. Regardless of where the exaggeration happens, a result seems more realistic if you can explain why you think it is happening.

Corresponding author
Leslie Myint, lmyint@macalester.edu

Most researchers do not deliberately claim causal results in an observational study. But do we lead our readers to draw a causal conclusion unintentionally by explaining why significant correlations and relationships may exist? Once we discover that an association exists, it is natural to want to explain why it does. We may describe potential mechanisms, make connections to previous literature, or put an observation in context. Despite these explanations, causal relationships are not proven in a single observational study and are only increasingly substantiated over the course of many such studies. There is observational evidence suggesting a noticeable prevalence of inappropriate causal language in both nutritional (*Cofield, Corona & Allison, 2010*) and educational (*Robinson et al., 2007*) research studies.

The distinction between correlational and causal evidence is not merely a pedantic formality. Because causal statements carry moral underpinnings, they can have dangerous consequences for societal perceptions of certain groups, products, or practices when consumed and interpreted by the general public (*Lombrozo, 2017*). For example, researchers of developmental origins of health and disease published a cautionary commentary in response to a collection of headlines (Mother's diet during pregnancy alters baby's DNA, Pregnant 9/11 survivors transmitted trauma to their children) that seemed to vilify mothers for developmental outcomes in babies (*Richardson et al., 2014*). In research areas dealing with human subjects, mistakes in perceptions about evidence can be harmful, and reporters must use great care in the language they use to describe scientific findings. The danger in these headlines and in related causal language (e.g., explanatory statements, jargon) lies not in the words themselves but in their interpretation by the public.

In this work, we investigate how interpretation of scientific evidence is affected by a specific area of causal language: explanation. We report the results of a randomized experiment performed on an online educational platform that suggest a strong effect of explanatory language on students' perception of whether a study is correlational or causal. We emphasize that the results presented here pertain to a single example (an observational smoking-cancer study) and that there are many opportunities for further research to clarify the extent of generalizability of these results.

## METHODS

Different types of studies have different analysis goals (Table 1) (*Leek & Peng, 2015*). We were interested in whether people can distinguish between a study whose goal was inferential and one whose goal was actually causal, as this is a common error often termed "correlation does not equal causation". We wanted to know whether including language explaining an observed association leads people to believe that an inferential study is causal. To test this hypothesis, we considered an experiment in a large online open-access data analysis course, for which one of the authors was an instructor. The other two authors had no relationship with the online course, and none of the authors had access to the participants' identities. This introductory-level course covered basic data analytic concepts. Our experiment involved a single randomized quiz question administered during the course. This experiment originally ran in January 2013 and was later independently replicated in a separate offering of the

**Table 1** **Goals for different analysis types (*Leek & Peng, 2015*).** These analysis types form the set of possible answer choices in our randomized experiment and were taught to students before the experiment was performed.

| Type of analysis | Goal of analysis |
| --- | --- |
| Descriptive | Summarizing the data without interpretation |
| Exploratory | Summarizing the data with interpretation, but without generalization beyond the original sample |
| Inferential | Generalizing beyond the original sample, with the goal of describing an association in a larger population |
| Predictive | Generalizing beyond the original sample, with the goal of predicting a measurement for a new individual |
| Causal | Generalizing beyond the original sample, with the goal of learning how changing the average of one measurement affects, on average, another measurement |
| Mechanistic | Generalizing beyond the original sample, with the goal of learning how changing one measurement deterministically affects another variable's measurement. |

course in October 2013. Between these two replications, over 22,000 students completed versions of our experimental question. These experimental questions were initially included as part of a regular quiz within the course; as such, we did not prospectively obtain informed consent. We later sought and were granted IRB approval to perform the analyses presented here (Johns Hopkins Bloomberg School of Public Health IRB number 00005988.) Consent was waived on the grounds that this retrospective analysis of the data was of a low-risk educational type and that we would remove any individualized identifiers in the data.

Early in the course, students were presented with the definitions of six possible types of data analysis (descriptive, exploratory, inferential, predictive, causal, and mechanistic) consistent with those shown in Table 1. In the subsequent course quiz, we provided students with a description of an inferential study - from which we can only infer correlation:

> We take a random sample of individuals in a population and identify whether they smoke and if they have cancer. We observe that there is a strong relationship between whether a person in the sample smoked or not and whether they have lung cancer. We claim that the smoking is related to lung cancer in the larger population.

We randomized students to see or not see an explanatory interpretation accompanying this description. Students in this explanatory interpretation group saw an additional sentence:

> We explain we think that the reason for this relationship is because cigarette smoke contains known carcinogens such as arsenic and benzene, which make cells in the lungs become cancerous.

All students were then asked to identify the type of analysis for these results. In addition to the correct answer (inferential), students were presented at random with three of four possible incorrect answer choices (descriptive, causal, predictive, mechanistic). That is, approximately 25% of students made their choice from inferential, descriptive, causal, and

**Table 2 Effect of explanatory language on student responses.** For each of four sets of answer choices seen, differences in the percentage choosing the "causal" and "inferential" answer choices are given, as well as 95% confidence intervals for the differences and sample sizes.

| Answer choices seen | Difference in percentage choosing "causal" when seeing explanatory language vs. not seeing explanatory language (95% CI for difference in proportions) | |
| --- | --- | --- |
| | January 2013 course | October 2013 course |
| Inferential, causal, descriptive, predictive | 14.5% (12.2%, 16.8%) $N = 5{,}061$ | 14.3% (6.4%, 22.2%) $N = 447$ |
| Inferential, causal, descriptive, mechanistic | 15.8% (13.4%, 18.1%) $N = 5{,}092$ | 14.8% (6.6%, 23.0%) $N = 463$ |
| Inferential, causal, predictive, mechanistic | 15.2% (12.8%, 17.5%) $N = 5{,}088$ | 19.9% (11.5%, 28.3%) $N = 437$ |
| | Difference in percentage choosing "inferential" when seeing explanatory language vs. not seeing explanatory language (95% CI for difference in proportions) | |
| Inferential, descriptive, predictive, mechanistic | −7.3% (−9.3%, −5.2%) $N = 5{,}016$ | −4.9% (−12.6%, 2.9%) $N = 416$ |

predictive, approximately 25% from inferential, descriptive, causal, and mechanistic, and so on. That only four answer choices could be shown at the same time was a limitation of the online platform. Although the described analysis is inferential in nature, we hypothesized that students who saw the explanatory language would be more likely to identify the analysis as causal if given that choice. Because students were able to retake this quiz multiple times in order to achieve a passing grade, we collected answers from each student's first attempt. We compared the proportions of the causal and inferential answer choices between the explanatory language and non-explanatory language arms using two sample tests for differences in proportions (Table 2).

## RESULTS

In our original experiment (January 2013), 20,257 students completed our experimental quiz question. These students were randomly assigned to one of four arms, where each arm contained the correct answer choice (inferential) and three incorrect answer choices (from among causal, descriptive, predictive, and mechanistic). Sample sizes are given in Table 2. We present detailed results for two arms: (1) students who chose between inferential, causal, predictive, and mechanistic analyses and (2) students who were not given causal as a choice, but instead chose between inferential, descriptive, predictive, and mechanistic analyses. Detailed results for the other student groups can be found in the Supplemental Information. Table 2 shows summary results for the four groups of students corresponding to the four sets of answer choices seen.

Among students selecting from inferential, causal, predictive, and mechanistic answer choices, the majority (68.5%) correctly answered that the description referred to an inferential data analysis (Table 3). However, a significantly higher percentage of students who were shown the explanatory language claimed it was a causal analysis compared to students who did not see the additional language: 31.8% compared to 16.6% (Difference in proportions: 15.2%. 95% CI for difference in two proportions: 12.8%–17.5%) (Table 2).

**Table 3  Detailed results for the experimental arm with answer choices: inferential, causal, predictive, and mechanistic.** In the presence of explanatory language, nearly twice as many students incorrectly selected "causal" with a corresponding decrease in the percentage of students correctly selecting "inferential".

| This is an example of a/an ________ data analysis. | January 2013 course (N = 5,088) | | October 2013 course (N = 437) | |
|---|---|---|---|---|
| | Saw explanatory language (N = 2,516) | No explanatory language (N = 2,572) | Saw explanatory language (N = 199) | No explanatory language (N = 238) |
| inferential | 1,508 (59.9%) | 1,977 (76.9%) | 116 (58.3%) | 190 (79.8%) |
| causal | 799 (31.8%) | 427 (16.6%) | 68 (34.2%) | 34 (14.3%) |
| predictive | 120 (4.8%) | 138 (5.4%) | 8 (4.0%) | 11 (4.6%) |
| mechanistic | 89 (3.5%) | 30 (1.2%) | 7 (3.5%) | 3 (1.3%) |

These results indicate that explanatory language increases the chance a student will mistake an inferential result as causal. In this case, students who saw the additional explanation were almost twice as likely to claim the results as causal.

This increase in the choice of a causal analysis when faced with explanatory language corresponded to a decrease in choice of an inferential analysis. The percentages of students who chose either a predictive or descriptive analysis were similar between the two treatment groups. However, there was an increase in the percentage of students who claimed the result was mechanistic in the explanatory language group: 3.5% compared to 1.2% (Difference in proportions: 2.3%. 95% CI for difference in two proportions: 1.5%–3.2%). This is not surprising since a mechanistic result is similar to a causal result in that it describes a deterministic process by which one variable affects another.

Among students who were not given the option to select "causal" as an answer (selecting instead from inferential, predictive, descriptive, and mechanistic analyses), a higher percentage (84.6%) correctly answered that the description referred to an inferential data analysis (Table 4). In this case, a significantly higher percentage of students correctly claimed the analysis was inferential when not shown the explanatory language: 88.2% compared to 80.9% (Difference in proportions: 7.3%. 95% CI for difference in two proportions: 5.2%–9.3%) (Table 2). These results indicate that, even without the ability to identify the analysis as causal, students had a harder time correctly identifying an inferential study when given hypothesized information about the reason for a correlation. The size of the effect is much smaller than with the causal answer option, however. The decrease in correct answers again corresponded to an increase in choice of a mechanistic analysis: 5.6% compared to 1.4% (Difference in proportions: 4.2%. 95% CI for difference in two proportions: 3.2%–5.3%).

To confirm our results, we analyzed data from an independent replication of our experiment in a later offering of the same data analysis course. In the replication (October 2013), 1,762 students completed our experimental quiz question. The results of this replication were consistent with those in the original experiment (Tables 2–4, and Supplemental Information). Differences in percentages for the causal and inferential answer choices were always of the same sign between the two courses, and the magnitudes of the differences were also similar (Table 2). While the sample size in this course is much smaller, the concordance of results and the maintenance of experimental procedures

**Table 4** **Detailed results for the experimental arm with answer choices: inferential, descriptive, predictive, and mechanistic (no causal).** In the presence of explanatory language, a lower percentage of students correctly selected "inferential", and a higher percentage of students incorrectly selected "mechanistic".

| This is an example of a/an _________ data analysis. | January 2013 course (N = 5,016) | | October 2013 course (N = 416) | |
|---|---|---|---|---|
| | Saw explanatory language (N = 2,485) | No explanatory language (N = 2,531) | Saw explanatory language (N = 199) | No explanatory language (N = 217) |
| inferential | 2,011 (80.9%) | 2,232 (88.2%) | 160 (80.4%) | 185 (85.3%) |
| predictive | 196 (7.9%) | 181 (7.2%) | 10 (5.0%) | 12 (5.5%) |
| descriptive | 138 (5.6%) | 82 (3.2%) | 14 (7.0%) | 14 (6.5%) |
| mechanistic | 140 (5.6%) | 36 (1.4%) | 15 (7.5%) | 6 (2.8%) |

between courses align with a statistical definition of replicability that has been put forth (*Patil, Peng & Leek, 2016*).

## CONCLUSIONS

We know that the way data is visualized can affect how well people derive information from graphs (*Cleveland & McGill, 1985*). The results of this experiment suggest that the way we write about a data analysis may also be critical. By performing a randomized controlled trial, we have shown clear evidence of an effect of explanatory statements on perceptions of research results for a specific scenario and replicated this effect in a second experiment. The nature of our study design justifies the use of causal language to describe the precise effect of explanatory language on categorical perceptions of research findings, but it is important to keep in mind that these effects are specific to a certain population of learners and to our specific quiz question. Further, our study does not elucidate any of the latent psychological or cognitive mechanisms that give rise to the shift in categorical perceptions. In the remainder of this section, we discuss these limitations and avenues for further research.

One limitation of our study is the population of participants. We performed this randomized trial in a study population of learners in a massive open online course (MOOC) as opposed to a representative sample of the target population of English speakers. We do not have access to demographic information on the learners in our trial because the online platform on which we conducted this study did not collect such information at the time. The composition of students in MOOCs has changed rapidly since then, so we are also unable infer demographics confidently. Some general research into MOOC demographics has been conducted; surveys indicate that these learners are slightly more likely to be male, often have bachelor's degrees, and typically have some level of employment (*Bayeck, 2016*). Learners in these online courses report a variety of motivations for taking the courses, suggesting at least some lifestyle diversity. However, there is certainly a gap between our MOOC study population and our target population, and we advocate rectifying this gap with collection of pertinent demographic information in future studies.

A second limitation of our study was the choice to use smoking and cancer as the scenario in our quiz question. Belief bias describes the tendency of individuals to judge the strength of a statistical claim based on its plausibility rather than on the data and methods

themselves, and it is likely to be operating in this study. The causal link between smoking and lung cancer has been firmly established over time through an accumulation of evidence, so although the wording of our quiz question does not describe a causal study, belief bias likely nudges learners to think otherwise. Had we used a different example, the effect of the explanatory text would likely have been smaller. We urge future studies to use examples that span a wider range of evidence for a causal relationship (e.g., the smoking example used here and one involving the link between yoga and anxiety disorders). As we discuss later, this will be key to improving our understanding about when exactly explanation contributes to causal misinterpretations.

There are specific avenues of further research that would elucidate the situations in which the presence of explanation influences the formulation of causal interpretations. As we have just discussed, the example of smoking and cancer is a difficult one for learners to contend with because the causal relationship has actually very well substantiated over time. Further, the wording of the explanatory text is pretty forceful in that it describes arsenic and benzene as being ''known carcinogens, which make cells in the lungs become cancerous''. Despite the strong wording of the statement, it is still unclear if the amount of these compounds in cigarettes is sufficient to be carcinogenic. This is a subtle point and further increases the difficulty of this question for students who are randomized to see the explanation. Future experiments should vary the persuasiveness of the explanatory text from being hypothetical but convincing to being far-fetched. The degree of persuasiveness of the explanatory text and additional nuances of how the explanation is conveyed might have an impact on specific cognitive mechanisms leading to causal interpretation. These additional nuances might include providing citations for claims, varying the level of detail in the explanation, or making known additional explanations that can be substantiated or discredited. If future experiments reveal differences in these modes of presentation, we could much better understand why misinterpretations occur.

Another limitation that is somewhat orthogonal to our use of specific features in the explanation text is the span of the scientific domain covered by our quiz question. The particular example we used falls within the field of medical biology. Future experiments should represent a diverse set of fields including social science, political science, economics, medicine, and biology subfields. It is possible that explanations with the same features carry different weight across fields. For example, it could be that biological explanations have greater cognitive impact because they are rooted in physical principles. Experiments that cover interactions between the field explored and the explanation provided have the potential to inform specific guidelines for reporting in different domains.

Our study has focused on the nature of causal misinterpretation in the presence of explanatory language. We have not investigated any strategies for combating these misinterpretations, an area of future research that could lead to actionable interventions regarding scientific communication. We recognize that it is quite difficult to avoid any explanation when communicating scientific results because explanation is a key means of interpreting research findings. Interpretation is essential for combining different sources of information and advancing our understanding. In both academic and mainstream scientific writing, there is a desire to put results into context, including hypothesized mechanistic

explanations to enhance the narrative around a set of empirical results. Nearly every study includes this type of explanation in the discussion section. However, our results suggest that such efforts may actually cause a certain population of readers to be misled about the strength of the scientific evidence. The misinterpretation may be exacerbated by the phenomenon that readers are swayed to believe a statement when they are told scientists understand it (*Sloman & Rabb, 2016*). Because interpretation, and thus explanation, is an essential aspect of science communication, we should not aim to avoid explanation but to understand how certain characteristics of explanation help or hinder perception. We hypothesize that it may be beneficial for readers' perceptions to follow up any explanations with warnings against interpreting results causally. Further research is needed to determine if this could counteract the effect of explanations on causal perceptions.

Because our study explores the nature of causal misinterpretation, it is also worth discussing different definitions and evaluations of causality and how they relate to the idea of causal misinterpretation as a whole. The Bradford Hill criteria make up one such evaluation tool. Two of the nine considerations outlined in the set of criteria, plausibility and coherence, relate to the results presented here. Plausibility refers to the presence of a realistic mechanistic explanation for an observed association, and coherence refers to an observed association that does not violate well accepted current knowledge. Evaluations of both plausibility and coherence are at play when students read the explanation about carcinogens. Given that modern day science applies these criteria in many different application areas (e.g., vitamin D levels and breast cancer risk (*Mohr et al., 2012*), Zika infection and birth defects (*Rasmussen et al., 2016*)), why do we balk when students make causal interpretations for the smoking example in our study?

One answer is that the Bradford Hill criteria are really just a collection of considerations and do not provide guaranteed evidence for a causal relationship. Thus the presence of a plausible and coherent explanation for the smoking-cancer association is pleasing but not definitive. Furthermore, the application of the criteria in a given situation can be problematic if understanding of the system under study is poor (*Höfler, 2005*). This can result from undue reliance on one's prior beliefs or from high uncertainty in the knowledge base on which one's prior beliefs are derived.

A second reason for our discomfort with the causal categorization of the smoking-cancer analysis is that we have a certain definition of causality in mind that the quiz question text simply does not support. The definition of a causal analysis presented in Table 1 describes the goal as describing (quantifying) how one variable changes on average when another variable is changed. Implicit in this definition is the ability to manipulate the intervention variable, and this ability to manipulate is not an absolute necessity in other forms of causality that have been discussed in other areas, such as the social sciences. Also implicit in this definition is the idea of ceteris paribus; all other variables remaining equal, what is the effect of manipulating the intervention variable, on average? For both of these reasons, it is important for further research to dissect more precisely the understanding of the word "cause" that is triggered when students see explanation accompanying the presentation of a statistical association. In the social sciences, there is a concept of sufficient motive causality that describes how belief in a causal relationship can be substantiated by

explaining why a sufficient set of motives could have led to a result (*Tellings, 2017*). For example, this type of reasoning is used by historians to understand why certain events occurred. Although sufficient motive causality is at odds with the definition we have used in this study, we do see a reconciliation. Sufficient motive causality is used in the social sciences because practitioners can assess the quality of evidence that it provides and because awareness of sufficient motives for an effect is sometimes all that is needed to elicit some other desired action. Both of these reasons are quite justifiable in that both acknowledge the role of explanation in causal reasoning as the Bradford Hill criteria also do, but they also make clear the consequences of buying in to such thinking. Assessing the quality of a sufficient motive explanation makes clear that an explanation for a result does not provide guaranteed actionable answers as if through a clear physical mechanism, a misconception known as statistical reification (*Greenland, 2017*). In assessing the quality of an explanation, practitioners should be mindful of various factors that can influence their judgment: strength of prior beliefs, populations for which the explanation holds, and details of statistical methods used to generate the explanation (*Colombo, Bucher & Sprenger, 2017*). If an individual consciously thinks about these matters, there is at least a healthy degree of transparency about the explanation. Using explanations to motivate desired actions also involves transparency regarding the nature of the explanation. For example, if a therapist knows that a treatment regimen works for her patient but not how, it is still worthwhile to convey an explanation for how the treatment works to motivate her patient to continue being treated. In a sense, the construction of this explanation is a causal analysis of the situation.

It could be that students in this study are increasingly categorizing the smoking-cancer study as a causal analysis in the presence of explanation because they are using reasoning similar to sufficient motive causality as described above. In this case they are perhaps not wrong in the grander debate about defining causal analysis. However we still have concerns that relate to the second implicit part of our definition of a causal analysis: the idea of ceteris paribus. Comparing individuals who smoke to those who do not smoke, who are otherwise identical, is there truly an increase in the proportion who develop lung cancer? Our worry is that explanation that accompanies an inferential result might lead individuals to forget this idea and not think carefully about confounding. Future experiments should carefully use wording to elicit specific concepts of causality such that participant responses more directly indicate thought processes.

In conclusion, we have shown a clear effect of explanations on categorical perceptions of statistical results in a specific scenario. However, it remains unclear what this implies about cognitive processes or actions that may result from interpretations. In the discussion above, we have highlighted specific avenues of further research that would better elucidate when, why, and how changes in interpretation follow from explanatory language. These avenues include broadening the populations studied, diversifying the subject domains covered, varying features of the explanatory language, and isolating different components of the definitions of causality to understand subsequent intentions. The last of these will be the most difficult but arguably the most important because it could shed light on what people might actually do in response to consuming a statistical result with explanation, whatever

the semantics of causation might be. For this, allowing more flexibility in responses (e.g., free text) might better facilitate capturing these thoughts. With future research, we hope that the community can reach a level of understanding regarding the effects of explanation that engenders vastly improved scientific communication.

### Funding
This work was supported by a National Institutes of Health grant (R01GM115440). The funders had no role in study design, data collection and analysis, decision to publish, or preparation of the manuscript.

### Grant Disclosures
The following grant information was disclosed by the authors:
National Institutes of Health: R01GM115440.

### Competing Interests
The authors declare there are no competing interests.

### Author Contributions
- Leslie Myint and Leah R. Jager analyzed the data, contributed reagents/materials/analysis tools, prepared figures and/or tables, authored or reviewed drafts of the paper, approved the final draft.
- Jeffrey T. Leek conceived and designed the experiments, performed the experiments, contributed reagents/materials/analysis tools, authored or reviewed drafts of the paper, approved the final draft.

### Human Ethics
The following information was supplied relating to ethical approvals (i.e., approving body and any reference numbers):

We received approval to analyze this data from the Johns Hopkins Bloomberg School of Public Health: IRB number 00005988.

### Data Availability
The code and data used to perform this analysis are available at GitHub: https://github.com/leekgroup/explanatory_language.

### Supplemental Information
Supplemental information for this article can be found online at http://dx.doi.org/10.7717/peerj.5597#supplemental-information.

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
