# Peer review of "Explanation of observational data engenders a causal belief about smoking and cancer"

_PeerJ, doi:10.7717/peerj.5597_

## Round 0.1 · original submission · Major Revisions

The reviewers have provided a range of views on your manuscript.

Reviewer #1 has raised some minor points, which you should be able to address without much trouble. Your relationship with the course would be an especially useful addition to the manuscript.

Reviewer #2 is similarly supportive and has raised a number of points, which again I would expect you to be able to address. Discussing the Bradford Hill criteria, and work on causality from psychology, would only strengthen the story presented here. Possible additional recent references to those provided by Reviewer #2 might include https://doi.org/10.1016/bs.plm.2016.11.006 https://doi.org/10.3389/fpsyg.2017.01430 and https://doi.org/10.1177/0959354317726876 but you could easily find dozens, if not hundreds, of recent articles and chapters to replace or add to these. I will leave the addition of specific references up to you but you need to develop the theoretical framework more and these or similar references would help you do this.

Reviewer #3 raises some challenging questions. In particular, their comments on the participant’s likely knowledge of the association between smoking and lung cancer (4) and their comment about the mechanism (5) need to be carefully addressed. Their comment 8 might be addressed by incorporating more of a framework for this study, including through the development of the background and further elaboration in the discussion. Similarly, as they note (10), even defining causality is problematic in itself. Again, this would provide an avenue to broaden the manuscript.

I really like this short manuscript and I could see it being a useful “conversation starter” for many, many researchers, but, at the moment, it is too short and narrow. Like Reviewer #3, I would like to see further studies of the question but I wouldn’t like to see this manuscript held up by that requirement. Along with boosting the background, adding a clearer plan for “further research” to the conclusions would also help to strengthen the manuscript's immediate impact.

·

Basic reporting

Thank you for the opportunity to review this manuscript. I had the opportunity to previously review this manuscript for another journal and the authors have addressed nearly all of my previously stated concerns. I summarise how these have been addressed and note a few minor outstanding comments.

The authors highlight the problem that the results of observational studies are frequently reported using causal language in the literature and media. They conduct a randomized controlled trial in the context of a massive online open course on data analysis to test the hypothesis that explaining an inferential analysis will cause readers to interpret the result as causal. The authors have made a strong case for why “it may be dangerous to explain” observational results. The nature of the “danger” is clearly articulated, and supported by salient examples.

A strength of this experiment is that it was replicated in the context of a second offering of the online course and the results were consistent. The manuscript is well-written and clear for a generalist audience, though I found that the Tables (particularly the titles), could be further streamlined/pared down to avoid repetition with the text.

The first example in the first sentence of the main text (“Facebook could raise your risk of cancer”) does not suggest a causal relationship to me as it uses the word “could” and “risk” where the other two are excellent examples. Could you replace the first with a stronger example?

The authors retrospectively sought and were granted IRB approval for their experiment. Since consent was not obtained prospectively, it might be important to note on what grounds a waiver of informed consent was granted. I would also be interested to know the relationship between the study authors and the course (e.g. were they instructors? Course designers?) and whether they had access to participants’ identities.

Experimental design

This is a novel experiment that provides a strong level of evidence to understand the impact of explanatory statements on readers' interpretation of inferential results.

Since the primary outcome was measured using only a single quiz question, the authors have appropriately highlighted the limitations of the study design in terms of measurement, the context in which results can be interpreted and suggested future research that would strengthen the causal claim that explanation accompanying an inferential result prompts causal interpretation.

The authors have helpfully provided some information about the sample enrolled in the MOOC, which will aid in generalizing these results. However, why was the course enrollment/sample size so much smaller in the replication (October 2013 results)? That said, the authors have adequately justified why the smaller replication sample is unlikely to bias the replication results.

Validity of the findings

The results are robust, statistically sound and the design is of low risk of bias.

The authors have provided a useful discussion of the role of explanation and interpretation in scientific writing and suggested future work that would reduce bias in reporting.

·

Basic reporting

The paper is well-written. I only miss some points and discussion (see below)

Experimental design

Investigating the impact of reporting a (substantive) explanation of an association found seems to be closely related to Bradford Hill's considerations of "plausibility" and "coherence". Maybe, the explanation for the effect described by the authors is that researchers use this heuristics. The authors might want to comment on when this really increased the probability that the causal conclusion is valid*.

*
Höfler M. The Bradford Hill considerations on causality: a counterfactual perspective. Emerging Themes in Epidemiology 2005; 2:11

Similarly as the authors, I suspect that most of the study participants yet take the example of the association between smoking and cancer as a study with a causal aim - because this has been debated for many decades, and it has been obvious for long that there truly exists a (strong) causal effect. However, the experiment is valid anyway because it is randomized, wherefore difference in outcomes can be attributed to the different further information provided. Surprisingly for me, the base rate is surprisingly low for this example. The low base rate, on the other hand, indicates validity. But I don't understand why the authors assume a smaller effect in a larger study (discussion).

Validity of the findings

I would not use the term "prove" because, strictly speaking, a causal effect can never be fully proven.

A description of the student sample is required (beyond the short discussion part on this). At least this has to provide basic socio-demographical information. Ideally, the study has collected any variables assumed to modify the effect under consideration. Then their distribution in the sample inform on possible direction and amiunt of selection bias when inferring on whatever target population (remains unclear here).

Besides, the authors need to report which statistical methods they have used.

Additional comments

I consider experimental studies on the impact of reporting study results on the correctness of conclusions very promising (personally, I also consider doing experiments on that). This is because it has been found that using better statistical methods and reporting (confidence intervals, even Bayesian bias models) does not solve the basic cognitive problems*.

Especially, when a causal conclusion is done or subtly suggested, it remains often unclear on which assumptions this is based - leaving the reader unable to decide whether to follow the logical path of argumentation and, thus, a conclusion or not. Instead, I (also) suspect, the decision is based on hidden heuristics. Accordingly, this experiment investigates a mechanism that is not pretty obvious but might be relevant: whether a substantive explanation is suggested.

*
Greenland S. Invited Commentary: The Need for Cognitive Science in Methodology. Am J Epidemiol. 2017186(6):639-645

**
Höfler M, Venz J, Trautmann S, Miller R.. Writing a discussion section: how to integrate substantive and statistical expertise. BMC Med Res Methodol. 2018 Apr 17;18:34.

·

Basic reporting

The manuscript is well written and is easy to follow. The authors’ hypothesis is clearly defined. The analyses and results are easy to understand and interpret.

1) line 18: “includes explanation” should be “includes an explanation”

2) (very minor): in analysis_FINAL.Rmd, line 269 says “causal” but it should be “inferential”

Experimental design

The authors test the hypothesis that providing a causal explanation for an observational study makes it more likely that readers will believe a “causal analysis” was used for the study. After describing an observational study where smoking is associated with lung cancer, participants were asked to choose which type of analysis was used for the study. Four of five types of analysis were given as choices (inferential, descriptive, predictive, mechanistic, and causal) in a multiple-choice format.

3) It’s not clear why the authors didn’t present all five choices to each participant. I made the assumption that this was a technical limitation with the online course that students were participating in, but the reasoning for this should be made explicit. Because of this design, the authors had to repeat the same analysis four times, and only one analysis was presented in the manuscript (the rest are in the supplementary material).

Validity of the findings

The authors find that when given a causal explanation for the results, participants are much more likely to select “causal analysis” and less likely to select “inferential analysis.” I have major concerns about whether these results would extrapolate to other analyses and explanations.

4) My biggest concern is that the explanation given to participants is true: smoking does cause lung cancer. I suspect this has a large influence on participant’s belief about the study’s design. The authors allude to this in the Conclusions and call this an example of the availability heuristic, though I believe it is better explained by belief bias.

5) The authors state that researchers often include “hypothesized mechanistic explanations,” but the link between smoking and cancer is not presented as a hypothetical explanation. It is presented as a true fact: they say “cigarette smoke contains known carcinogens,” which is tantamount to saying “cigarettes are known to cause cancer.” This is exactly what the observational study is supposedly testing.

6) Participants are asked to identify “the type of analysis for this results.” Is it clear to participants that the explanation is not part of “the analysis”? To non-scientists, the distinction may not be clear; we often use the word “analyze” in common vernacular to mean “explain” (e.g., “analyze the situation”). If participants consider the explanation part of the analysis, they are not wrong for judging it to be causal. The experimental design should make this clear, e.g., by clearly labeling “ANALYSIS” and “EXPLANATION OF THE ANALYSIS”.

Additional comments

7) The authors test their hypothesis using only a single example (smoking and cancer). This is not sufficient evidence to support their broader hypothesis. The authors need to expand their materials and present many more examples/explanations that vary in different ways.

8) No mechanism is hypothesized for why participants draw an erroneous conclusion. Would the same results be observed if the explanation provided were non-causal? What if the explanation were presented as conjecture, and not fact? Is this inference affected by participants’ prior beliefs? Does it hold only in biological domains, or would it work for social science (psychology/economics) explanations as well? The authors need to diversify their materials and test a range of possible scenarios to determine where and why this effect occurs.

9) The authors fail to report statistical tests throughout the manuscript, though their R code (supplementary material) shows they performed these analyses (chi-square tests). These should be presented whenever comparisons are made in the manuscript.

10) The authors provide a definition of “causal analysis”, as well as other types of analysis. I don’t have any major qualms with the definition, but exact definitions of causality are debated in the psychological and philosophical literature. Would you expect the same results if participants were instead given examples of each type of data analysis? If the results differ, this might suggest that participants are more likely to believe the study is causal in part because they don’t understand what a causal analysis is (as opposed to mis-understanding the experimental design).

11) In the abstract, the authors say that their “results suggest that […] qualification of explanations could be a useful avenue of exploration to counteract the problem.” This is never actually tested.

I think the authors have an interesting and reliable effect in their data, and their research question is broadly worth pursuing. It should be explored more fully with additional studies.

---

## Round 0.2 · Minor Revisions

I think that your additions to the Discussion in particular have greatly strengthened the manuscript. One of the reviewers had no further comments but Reviewer 3 has provided further comments around the scope of the manuscript. While adding more experiments remains a possibility, if this was not done, I like their suggestion for the title, which would make it clearer that this is conversation starter rather than ender from the outset. Their suggestion about making this particular case/example explicit in the abstract would further help ensure that the scope of the present manuscript is easily recognised and appreciated. A similar clarification added to the final paragraph in the introduction would, I think, be sufficient to delimit what is shown here from the more general question so that this study motivates replication in other settings and with other questions.

A minor point is that when you present 95% CIs for differences in proportions in the text, you could, as a kindness to the reader to save them turning to the Table or grabbing their calculator, also present the actual difference in the text (Lines 132, 141, 149, 155).

As the manuscript is about a single case at the present time, perhaps Line 172’s “is also” would be better as a “may also be” (I think this is easier than writing a stronger version which refers to your particular question—details of which follow on Lines 174–177). Related to this, on Line 173, perhaps “clear evidence of an effect from… for a specific scenario”. For convenience, these two edits would make Lines 172–174 something like:

"The results of this experiment suggest that the way we write about a data analysis may also be critical. By performing a randomized controlled trial, we have shown clear evidence of an effect from explanatory statements on perceptions of research results for a specific scenario and replicated this effect in a second experiment."

A similar point arises at Lines 315–316, where again I think making it clear that this “clear effect” is for a single example would be worthwhile.

I wonder if at Line 203 you might want to give a similar-styled suggestion as you do for the first limitation (Lines 192–194), for example “and future studies could potentially use questions with a range of evidence for a causal relationship (e.g., the smoking example used here and one involving meat consumption and dental caries)”. Obviously the example is merely a suggestion if you decided to add something here. You do address such issues in much more detail later from Line 213 onwards, so I’m mostly suggesting some foreshadowing here and to mirror the preceding paragraph.

Lines 232–233 read like a limitation, but I feel that this is more a further area for research. Identifying such biases, starting with a single case here and then potentially being replicated in different settings and areas, would seem to me to be necessary before considering intervention strategies.

·

Basic reporting

None

Experimental design

None

Validity of the findings

None

Additional comments

None

·

Basic reporting

(minor) Sloman & Rabb citation is listed twice in the references

Experimental design

No comment

Validity of the findings

No comment

Additional comments

I thank the authors for considering my suggestions and those of other reviews. In particular, I appreciate the inclusion of statistical comparisons, the expanded discussion on the definition of causality, and a more nuanced discussion of the particular smoking example used.

However, I still believe that the singular “smoking” example used in the experiment is not sufficient to test the broader hypothesis that the authors contend (that “explaining an analysis will lead readers to interpret an inferential analysis as causal.”) To be clear, I agree with the authors that this specific example provides some evidence for that hypothesis (though limited), and I would like to see this manuscript published with additional stimuli that support this claim.

I am worried that what the Action Editor calls a “conversation starter” will actually be a “conversation ender.” Other researchers who take the time to test multiple stimuli more thoroughly may have their manuscript rejected on the grounds that the effect is no longer novel. If the authors plan on testing additional stimuli themselves, then why not include them in this manuscript?

The authors addressed four of my points at least partially in the rebuttal letter by a call for “future” research to resolve these issues. I agree, most of the issues I raised require additional studies. I should have been more clear in my initial review that I was suggesting the authors conduct an additional study to address some of these concerns. I realize it is common practice to assuage minor methodological disagreements through a padded discussion of the limitations of the current experiments. In this case, I don’t believe such discussion is sufficient.

If this manuscript is to be to published without additional studies, I would highly recommend making it very clear that the broader hypothesis was tested in a very limited scope, in both the abstract and the manuscript title (e.g., “Explanation of observational data engenders a causal belief that smoking and cancer”). Presently, the abstract does not mention that the hypothesis is tested with a single “smoking” example, which could give the false impression that the authors used a diverse set of stimuli (particularly because they conducted two massive experiments). I realize that this conveys a much more limited finding, but given that the effect has not yet been generalized beyond this singular example, I think it is warranted.

---

## Round 0.3 · Minor Revisions

Thank you for your revisions. There were no further comments from the reviewers and I am satisfied that you have struck a good balance between describing these interesting results, and so starting an important conversation, and motivating replication and extension of your findings.

I have worked through your manuscript carefully again and found a very small number of copyediting points which we should resolve now. Once that is done, I will be delighted to quickly recommend acceptance for this manuscript.

Line 30: I appreciate that citations of Internet pages remains challenging, but I think adding the year (2017) both here and in the references (Line 363) would be helpful to the reader. This then matches the approach used for other non-scholarly references (Lombrozo, 2017; Mirror, 2008; and PTI, 2016).

Line 35: While the reader ought to appreciate that this is a 95% CI, it would be worth making this explicit. I note that the reference does this for earlier results when reporting this but didn't repeat that information for all.

Line 385: a spurious blank line.

References: You're a little inconsistent with DOIs, mostly using the format "DOI: 12.3456/abc" but in a few cases using URLs (Lines 359, 361, 376, 384, and 400).

This is all that I can find, so congratulations on a well written and interesting manuscript! I look forward to formally accepting it very soon.

---

## Round 0.4 · accepted · Accept

Thank you for tidying up those last few points. I look forward to seeing how the conversation around this develops and perhaps to seeing further articles from your group on this interesting topic. Well done.

#